# Expanded Convolutional Neural Network Based Look-Up Tables for High Efficient Single-Image Super-Resolution

## ABSTRACT

Advanced mobile computing has led to a surge in the need for practical super-resolution (SR) techniques. The look-up table (LUT) based SR-LUT has pioneered a new avenue of research without needing hardware acceleration. Nevertheless, all preceding methods that drew inspiration from the SR-LUT framework invariably resort to interpolation and rotation techniques for diminishing the LUT size, thereby prolonging the inference time and contradicting the original objective of efficient SR. Recently, a study named EC-LUT proposed an expanded convolution method to avoid interpolation operations. However, the performance of EC-LUT regarding SR quality and LUT volume is unsatisfactory. To address these limitations, this paper proposes a novel expanded convolutional neural network (ECNN). Specifically, we further extend feature fusion to the feature channel dimension to enhance mapping ability. In addition, our approach reduces the number of single indexed pixels to just one, eliminating the need for rotation tricks and dramatically reducing the LUT size from the MB level to the KB level, thus improving cache hit rates. By leveraging these improvements, we can stack expanded convolutional layers to form an ECNN, with each layer convertible to LUTs during inference. Experiments show that our method improves the overall performance of the upper limit of LUT based methods. For example, under comparable SR quality conditions, our model achieves state-of-the-art performance in speed and LUT volume.

## CCS CONCEPTS

• **Computing methodologies** → **Computational photography**.

## KEYWORDS

image super-resolution; look-up table; expanded convolution

**ACM Reference Format:**
Anonymous Author(s). 2024. Expanded Convolutional Neural Network Based Look-Up Tables for High Efficient Single-Image Super-Resolution. In
. ACM, New York, NY, USA, 9 pages. https://doi.org/XXXXXXX.XXXXXXX

## 1 INTRODUCTION

Single-image super-resolution (SISR) aims to recover a high-resolution (HR) image with high-frequency image details from a single low-resolution (LR) image. Early methods for SR were based on interpolation and sparse coding, such as nearest neighbor, bilinear, bicubic

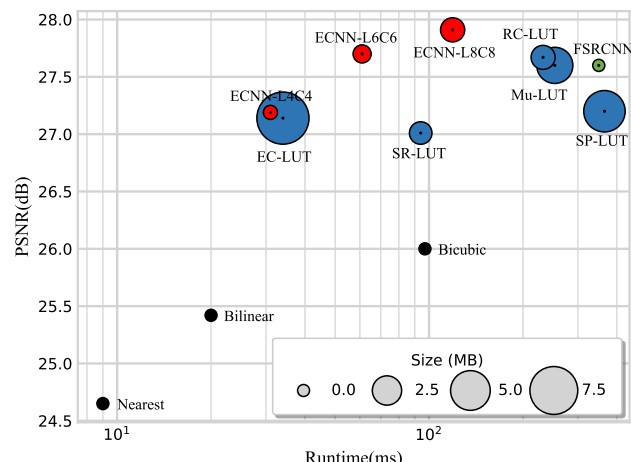

**Figure 1: Illustration of PSNR on Set14 benchmark dataset for ×4 SR, model size and runtime which measured by generating $1280 \times 720$ image. We compare our methods (red) with common interpolation based methods (black), prior LUT-based methods (blue) and deep learning based methods (green). Our models offer superior SR quality, faster inference, or smaller LUT volume compared to comparable LUT-based methods.**

[9], and A+ [27]. Interpolation-based SR methods often produce vague results. Sparse encoding methods, however, suffer from slow inference speeds. Recently, deep learning-based SR methods [11, 25, 17, 28] have achieved significant performance improvements. However, these methods have dramatically increased computational costs and even require large amounts of memory, hindering their practical applications on resource-limited edge devices like smartphones and smartwatches.

Look-up table (LUT) based methods store the outputs of complex computations in a LUT, which can be directly retrieved without recomputing when needed. The low computational cost and hardware-independent characteristics of LUT make it suitable for implementing efficient SR methods. However, obtaining a LUT requires enumerating the input values, which leads to an exponential growth of the LUT size with the increase of the index pixel number. When using a single pixel index, the LUT size is usually at the KB level. When using two pixel indices, the LUT size grows to the MB level. Although sampling techniques can be used to reduce the LUT size, on one hand, sampling techniques are limited in their ability to increase the index pixel number, typically only allowing for up to four indices. On the other hand, interpolation is required at the inference stage after sampling, which increases the computational costs. Cascading methods can make the LUT size grow linearly with the receptive field (RF) size, but these methods still remain

within the confines of the SR-LUT [8] framework and still require interpolation and rotation techniques.

Recently, a novel study [30] proposed an expanded convolution method to improve LUT-based SR methods' inference speed. By expanding the output window size in the spatial dimension thereby increasing the RF, using merely two pixels as indices, the performance outperforms SR-LUT that employs four pixels as indices. Meanwhile, due to avoidance of interpolation operations, it was much faster than SR-LUT. However, we found some problems in EC-LUT. First, the method still used rotation techniques without conferring any significant advantage. Second, it still used two pixels as indices, resulting in a LUT size of 9MB, which requires more storage space and reduces the cache hit rate. Finally, a bigger problem was that the expanded operation was performed in the HR space, significantly increasing the computational burden.

To address these issues, in this paper, we further improve the expanded convolution (EC) and generalize it to more common cases. At the same time, we completely abandon the SR-LUT framework and directly use EC as a vanilla convolution to build an expanded convolutional neural network (ECNN). Specifically, based on the previous version of EC, we further extend the expanded operation to the channel dimension while reducing the input window size to $1 \times 1$. Now, the input of EC is a single value, and the output size is $k_s \times k_s \times ch_{out}$, where $k_s$ denotes equivalent kernel size and $ch_{out}$ denotes the number of output channels. This is exactly opposite to the vanilla convolution. In this way, the LUT size becomes:

$$2^{b_1} \times k_s \times k_s \times ch_{out} \times b_2 \text{ bit} \tag{1}$$

where $b_1$ denotes bit width of feature map, $b_2$ denotes bit width of the value stored in LUT. We no longer need sampling, interpolation operations and rotation techniques while improving the cache hit rate. Moreover, we perform the EC operation in the LR space and transform to the HR space by directly splitting each value of the input features into $r \times r$ values. Finally, we aggregate these values along the channel axis to compress them back to the original low-dimensional RGB space. In summary, our contributions can be summarized as follows:

- We propose ECNN, a novel LUT-based approach for SR task. By reducing the number of single indexed pixels to just one, ECNN eliminates the interpolation and rotation tricks, thereby reducing the LUT size and breaking the inference speed bottleneck of the SR-LUT series methods.
- We further extend feature fusion to the feature channel dimension to enhance mapping ability. The smaller LUT size and multi-channel processing capability enable us to stack EC layers to construct ECNN, which significantly improves SR quality.
- Extensive experiments show that our method improves the overall performance of the upper limit of LUT based methods. For example, under comparable SR quality conditions, our model achieves state-of-the-art performance in terms of speed and LUT volume.

## 2 RELATED WORK

### 2.1 Traditional Super-Resolution

Interpolation-based methods are widely used, such as nearest-neighbor interpolation, bilinear interpolation, and bicubic [9] interpolation. These methods obtain SR results by taking the weighted average of pixels near the target location. Interpolation-based methods are simple and efficient. However, they only consider positional information and do not fully consider the arrangement of different pixel values, so they cannot effectively recover high-frequency signals. Sparse coding-based methods [27] infer HR images by learning the sparse representation of patches. However, computing sparse representations significantly increases inference time.

### 2.2 Deep Neural Network Based Super-Resolution

Deep neural networks (DNNs) have revolutionized the field of SR with their powerful fitting capabilities. Since the pioneering work of Dong et al. with SRCNN [5], the landscape of SISR has been enriched by a plethora of innovative models, each introducing novel architectural features and learning strategies [1, 6, 10, 11, 25, 13, 14, 15, 23]. The trend towards increasingly complex models [17, 28, 33] has led to significant performance gains. This is exemplified by the introduction of residual learning strategies and attention mechanisms [22, 32], which have become integral components of modern SR frameworks. The EDSR [17] and RDN [33] models, for instance, employ extensive residual blocks and have set new benchmarks in terms of image quality. Despite these advancements, the deployment of such models on edge devices remains a challenge due to their computational intensity and memory requirements. Techniques like quantization and pruning offer some relief by compressing the models without a substantial loss in performance. However, the need for specialized hardware like GPUs, DSPs, or NPUs for efficient inference underscores the gap between research prototypes and real-world applicability. Overall, while DNNs have significantly pushed the boundaries of SISR, the quest for practical, deployable solutions that balance performance with efficiency continues to be a driving force in the field.

### 2.3 LUT Based Super-Resolution

Look-up tables (LUTs) do not require special hardware implementation and only need some extra storage space to accelerate algorithm running speed by trading space for time. Jo et al. [8] were the first to apply LUTs to SR tasks and proposed SR-LUT. SR-LUT trains an SR network and then finds an equivalent LUT. The LUT replaces the complex DNN in the inference phase, significantly reducing the computational burden and making SR-LUT more practical. Subsequently, Li et al. [16] proposed MuLUT, which improved SR performance by using multiple LUTs to enlarge the receptive field (RF). Ma et al. [20] developed SP-LUT with a similar idea, using multiple cascaded LUTs to enlarge the RF. Note that SP-LUT quantized the intermediate features to 4 bits by using two parallel branches of MSB and LSB, thereby avoiding the interpolation operation. Meanwhile, they also discarded the rotation operation. This was an important attempt to break out of the SR-LUT framework. However, each module in SP-LUT only enlarges the RF by a small

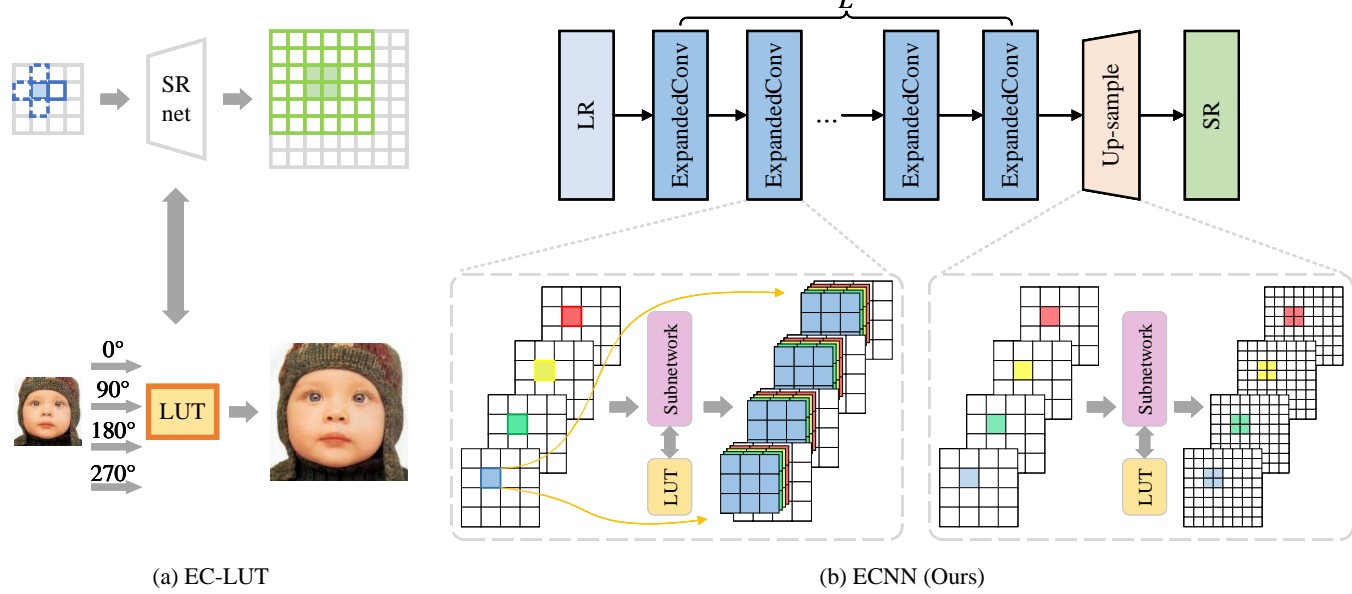

(a) EC-LUT                                                    (b) ECNN (Ours)

**Figure 2: Comparison of EC-LUT [30] and our ECNN method. The figure is illustrated for ×2 SR ($r = 2$). (a) The overview of EC-LUT. EC-LUT establishes the connection between LR and HR images using a single expanded convolution layer, while still retaining the rotation techniques of the SR-LUT series of methods. (b) Our ECNN architecture. The proposed ECNN is composed entirely of expanded convolution layers. The details of the EC layer within ECNN are depicted for hidden layers with four channels.**

part in the horizontal or vertical direction, and the method leads to the need to stack many modules to enlarge the RF, which seriously affects efficiency. Liu et al. [19] proposed an RC module, which significantly enlarged the RF and improved SR performance with only a little extra storage. However, RCLUT followed the SR-LUT framework, so it had no advantage in inference time. Recently, Yin et al. [30] introduced an expanded convolution method that increases the RF by expanding the output window size in the spatial dimension rather than the input window size. This novel approach has piqued our interest. Nonetheless, the performance of EC-LUT in terms of SR quality and LUT volume remains unsatisfactory.

## 3 METHOD

Figure 2 shows the overall workflow of ECNN and its differences from the EC-LUT [30] method. EC-LUT enlarged the RF by expanding the output window size of convolution in the spatial dimension, thereby maintaining the SR quality without interpolation and gaining an advantage in inference speed. To completely avoid the negative effects of interpolation and rotation operations, we improved the expanded convolution (EC) by further extending the expansion to the channel dimension and reducing the number of single indexed pixels to just one. Now, EC looks like an inverted vanilla convolution, which generates multiple output values from a single input value. The output values are placed in the corresponding positions and in-place addition is performed. To enhance the mapping ability, we construct an ECNN by connecting multiple EC layers.

### 3.1 Expanded Convolution

Conventional convolution operations compute the product of each pixel in an image with the corresponding elements of a filter in its neighborhood, followed by a summation to produce the convolution output. EC-LUT introduces an EC method that feeds pixels within the convolution input window into a neural network, which then outputs $n \times m$ values arranged into an output window. Each element of the output window is then added to the corresponding pixel in the output image. In other words, during the convolution process, data is read from the input image by the sliding input window and written into the output image by the output window (in-place addition operation). In EC, enlarging both the input and output windows can increase the RF. However, the size of the LUT is exponentially related to the input window size and linearly related to the output window size. Therefore, EC-LUT reduces the LUT size by enlarging the output window and shrinking the input window, whereas other LUT-based methods reduce the LUT size by sampling the LUT. Notably, the sampled LUT requires interpolation techniques during the inference stage to fill in missing dictionary values, significantly increasing computational complexity. It is precisely because interpolation is not needed that EC-LUT achieves faster inference speeds.

Following the concept of EC-LUT, we further reduce the EC input window to the size of a single pixel. At this point, the size of the LUT is significantly reduced from GB or MB levels to KB levels. The subsequent challenge is how to improve SR quality. Previous studies have improved SR quality by cascading multiple LUTs. However, we observed that even when cascading multiple

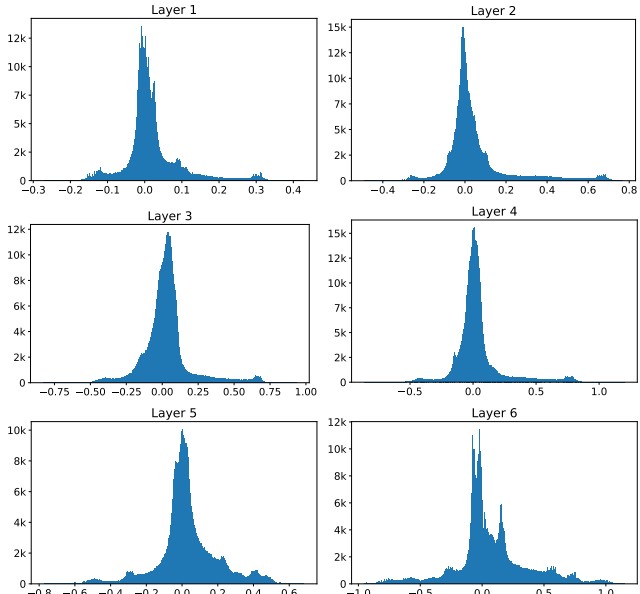

Query          Convertible LUT                    Value

**Figure 3: Details of sub-network in the EC layer.**

LUTs and employing different branches to process the input image separately, these methods are merely mapping one feature plane to another. In contrast, CNN networks first map the input image to a high-dimensional latent space and perform feature extraction in that space. Inspired by this, this paper further extends the EC feature fusion operation from the spatial dimension to the channel dimension. Furthermore, given the presence of multiple channels, it is logical to stack multiple EC layers to form an ECNN.

Figure 3 shows the details of sub-network in the EC layer. The sub-network consists of three $1 \times 1$ convolutions and two ReLU activation functions. To shorten training time, the hidden layer channel count of the sub-network is set to 64. The EC layer can be formally described as:

$$\mathbf{X}(i, j, c) = \Phi_\theta(\mathbf{F}_{in}(i, j, c)),$$
$$\mathbf{F}_{out}(i, j, c) = \sum_{x \in \chi} x(i, j, c) \tag{2}$$

where $(i, j, c)$ is the pixel index in the feature map, $\mathbf{F}_{in}(i, j, c)$ represents the input window at location $(i, j, c)$, $\mathbf{X}$ represents the corresponding three-dimensional sliding output window, $\Phi$ represents the learnable sub-network, $\theta$ represents the parameters of $\Phi$, $\mathbf{F}_{out}$ represents the final output feature obtained at the target position, and $\chi$ denotes the sliding window sets that cover the target position.

### 3.2 ECNN Architecture

As shown in Figure 2, EC can be applied as a regular convolution operation. Hence, our ECNN model consists solely of EC layers, and it extracts the feature maps in the LR space to lower the computational burden. Then, the model appends an up-sampling module at the final stage of the network to enhance the resolution. Furthermore, inspired by VGG [24], instead of using a single large-scale RC module to expand the receptive field as in RCLUT, we stack several $3 \times 3$ EC layers to increase efficiency.

The first layer of ECNN is responsible for ascending the input channel from $ch_{in}$ to $ch_{mid}$. The intermediate hidden layers use a $3 \times 3$ EC layer with input and output channels of $ch_{mid}$. Previous LUT-based SR methods have used multiple branches to enhance mapping capabilities, but the ends of these branches are always merged together. In other words, when these branches are considered as a whole, they always map one plane to another plane. However, on a single feature plane, the information contained in local patches is limited. ECNN uses multiple channels, which contain richer information in local patches, resulting in higher SR quality.

The final layer of ECNN is an up-sampling module, which directly splits each value of the input features into $r \times r$ values by

**Figure 4: After the convergence of the ECNN-L6C6 network without quantization nodes, the histogram of the output data for each layer on the Set5 dataset.**

non-linear mapping, as shown in Figure 2. Finally aggregates them along the channel axis to compress back to the low-dimensional RGB space. Actually, this procedure can also be viewed as setting the output window size of the EC layer to $r \times r$, and then when reordering the output, there is no overlap in the spatial dimension, and only overlap in the channel dimension.

It should be noted that another major distinction between ECNN and the SR-LUT framework is ECNN's ability to simultaneously process all three RGB channels. When there are ample channels in the feature maps extracted within the network, sharing a single set of extracted features for all three RGB channels can reduce computational requirements by approximately one-third, thereby further accelerating inference speed. However, when there are fewer intermediate feature channels, this shared feature approach may fail to adequately represent the information of all three RGB channels, compromising SR quality. This study continues to employ the approach of channel separation.

### 3.3 Quantization

Converting a neural network into a LUT necessitates computing the output values for all possible input values. These input values serve as indices for the LUT, with the corresponding output values stored at those indices. In the case of SR-LUT, the input is a LR image with a data bit-width of 8 bits, thus obviating the need for quantization operations. However, when cascading multiple LUTs, the intermediate feature tensors act as inputs to the hidden layers, which typically utilize 32-bit floating-point data during training. As shown in Equation 1, the LUT size is exponentially related to the input value's bit-width, it becomes necessary to quantize the feature maps.

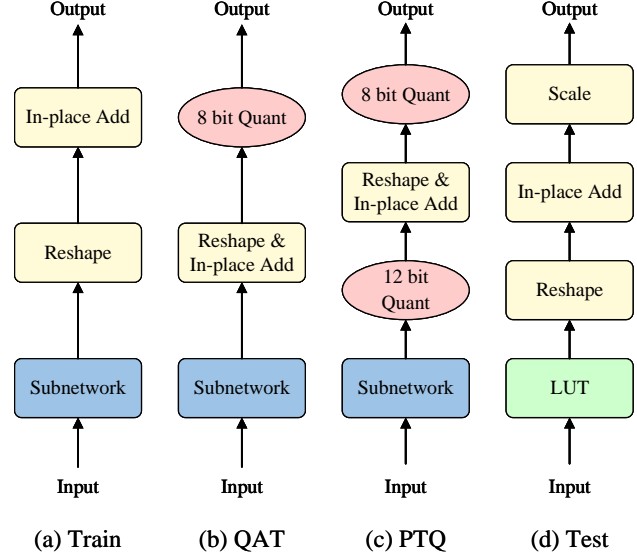

Output          Output          Output          Output

(a) Train     (b) QAT     (c) PTQ     (d) Test

**Figure 5: The various stages of transferring EC layer to LUT. (a) Normal training. (b) Fine-tuning with quantization node. (c) Zero cost post-training quantization. Traverse the input of the sub-network and store the 12 bits quantized value in LUT. (d) Replacing sub-network with LUT during test phase.**

Previous studies, such as SP-LUT, directly incorporate quantization nodes into the training process. However, feature maps produced by randomly initialized neural networks span a wide numerical range, and the truncation functions used in quantization nodes can impede gradient propagation. Experiments have demonstrated that this approach fails to achieve optimal performance. Alternatives like MuLUT and RCLUT employ Tanh or Sigmoid activation functions to map feature maps to a finite range. However, these methods still affect the model's convergence rate. To address this issue, we analyzed the results of standard network training, excluding quantization nodes. As illustrated in Figure 4, we observed that in a normally trained network, the output value range of each layer falls within $[-1, 1]$. Therefore, truncating the data to $[-1, 1]$ will no longer hinder gradient propagation. Based on this finding, we propose decoupling quantization from model training, as depicted in Figure 5. The model is first trained normally, and upon convergence, quantization nodes are introduced for fine-tuning to ensure the model's performance remains unaffected. This fine-tuning process is referred to as quantization-aware training (QAT).

The computation of the quantization node can be formulated as follows:

$$s = 2^{b-1} - 1,$$
$$Z = clamp(round(x \cdot s), -s, s), \quad (3)$$
$$y = \frac{Z}{s}$$

where $s$ is the scaling factor, which is determined by the quantization bit number $b$, $x$ is the input floating-point number, $Z$ is the integer after quantization, and $y$ is the output of quantization

node. We attempted to quantize the feature maps to 4 bits, but this resulted in a drastic performance drop, so we eventually opted for 8 bits quantization.

Note that QAT is performed on the input (LUT index) of the sub-network in the EC layer, whereas the output (values stored in LUT) of the sub-network remains a floating-point number prior to the in-place addition operation. In order to further reduce the LUT volume, the values stored in LUT were further quantified, but this step was postponed until after fine-tuning. For this step, experiments indicate that post-training quantization (PTQ) can be conducted directly, and quantization above 12 bits hardly affects performance. Thus, we quantize it to 12 bits at the LUT generation stage. It is worth noting that our PTQ does not require any statistical data about the model, which means that this step of quantization is zero-burden.

## 4 EXPERIMENT

### 4.1 Implementation Details

**Datasets and Metrics.** We use the DIV2K dataset [26] for training. This dataset has 800 images for training, 100 for validation, and 100 for testing. In addition, there are five commonly used benchmark test datasets, namely Set5 [3], Set14 [31], B100 [2], Urban100 [7], and Mang109 [21]. We report our results on these five datasets and compare them with previous studies. The quantitative evaluation metrics are PSNR (peak signal-to-noise ratio) on the Y channel of YCbCr space and structural similarity index (SSIM) [29]. In addition, We evaluate the computation efficiency by recording and presenting the rumtime of generating $1280 \times 720$ output images on mobile devices. To be consistent with previous studies, according to [17, 33], we use Matlab's imresize function to perform bicubic interpolation on HR images to obtain LR images.

**Training Details.** The sub-network within the EC layer uses a total of 3 grouped $1 \times 1$ convolution layers, with the number of groups set to $ch_{in}$, and the number of hidden layer channels set to $64 \times ch_{in}$, where $ch_{in}$ denotes the input channels of the EC layer. The sub-network in the up-sampling module of the last layer of ECNN adopt the same settings. We use Adam optimizer [12] with an initial learning rate of $1 \times 10^{-4}$ for a total of 20000 epochs, halving the learning rate every 4000 epochs. The loss function is mean-squared error (MSE). We randomly crop the LR image into patches of size $48 \times 48$ with a mini-batch size of 16 and augment the data by random rotation and flipping. We train the ECNN model with Pytorch [4] on Nvidia 2080Ti GPU.

### 4.2 Quantitative Comparison

We compared ECNN with other SR methods, including three common interpolation methods (nearest neighbor, bilinear, bicubic [9]), five LUT-based methods (SR-LUT [8], EC-LUT [30], MuLUT [16], SP-LUT [20], RCLUT [19]), and three neural network-based methods (FSRCNN [6], CARN-M [1], RRDB [28]). To compare the evaluation metrics of different dimensions more clearly, we implemented three different complexities of ECNN: ECNN-L4C4, ECNN-L6C6, ECNN-L8C8, where the number after 'L' indicates the number of EC layers, and the number after 'C' indicates the number of channels in the hidden layer.

**Table 1: Quantitative comparisons with other SR methods on 5 benchmark datasets for $r = 4$. The best values of LUT-based methods are shown in bold and the second-best values are shown in underline. Size denotes the storage space or the parameter number of each model. Runtime is measured on a MEIZU 16s smartphone for generating $1280 \times 720$ output images. $\sharp$ indicates that no runtime code was provided, and the runtime comes from their original paper.**

| Method | Runtime | Size | Set5 | Set14 | B100 | Urban100 | Manga109 |
|---|---|---|---|---|---|---|---|
| Nearest | 9ms | - | 26.25/0.7372 | 24.65/0.6529 | 25.03/0.6293 | 22.17/0.6154 | 23.45/0.7414 |
| Bilinear | 20ms | - | 27.55/0.7884 | 25.42/0.6792 | 25.54/0.6460 | 22.69/0.6346 | 24.21/0.7666 |
| Bicubic | 97ms | - | 28.42/0.8101 | 26.00/0.7023 | 25.96/0.6672 | 23.14/0.6574 | 24.91/0.7871 |
| SR-LUT [8] | 94ms | 1.274MB | 29.82/0.8478 | 27.01/0.7355 | 26.53/0.6953 | 24.02/0.6990 | 26.80/0.8380 |
| EC-LUT-V [30] | 41ms | 9MB | 29.91/0.8461 | 27.14/0.7419 | 26.61/0.7019 | 23.98/0.6977 | 26.96/0.8362 |
| EC-LUT-S [30] | 257ms | 11.466MB | 30.35/0.8592 | 27.45/0.7484 | 26.77/0.7062 | 24.28/0.7101 | 27.39/0.8466 |
| SP-LUT [20] | 365ms | 5.5MB | 30.01/0.8516 | 27.21/0.7427 | 26.67/0.7019 | 24.12/0.7058 | 27.00/0.8430 |
| MuLUT$^\sharp$ [16] | 253ms | 4.062MB | 30.60/0.8653 | 27.60/0.7541 | 26.86/0.7110 | 24.46/0.7194 | 27.90/0.8633 |
| RCLUT$^\sharp$ [19] | 232ms | 1.513MB | 30.72/0.8677 | 27.67/0.7577 | 26.95/0.7145 | 24.57/0.7253 | 28.05/0.8655 |
| ECNN-L4C4 (Ours) | **31ms** | **198.773KB** | 29.99/0.8524 | 27.19/0.7443 | 26.67/0.7034 | 24.06/0.7053 | 27.07/0.8460 |
| ECNN-L6C6 (Ours) | 61ms | 661.236KB | 30.75/0.8683 | 27.70/0.7583 | 26.96/0.7144 | 24.58/0.7271 | 28.12/0.8668 |
| ECNN-L8C8 (Ours) | 119ms | 1.543MB | **31.06/0.8753** | **27.91/0.7631** | **27.08/0.7180** | **24.82/0.7364** | **28.59/0.8762** |
| FSRCNN [6] | 371ms | 12K | 30.71/0.8656 | 27.60/0.7543 | 26.96/0.7129 | 24.61/0.7263 | 27.91/0.8587 |
| CARN-M [1] | 4955ms | 412K | 31.82/0.8898 | 28.29/0.7747 | 27.42/0.7305 | 25.62/0.7694 | 29.85/0.8993 |
| RRDB [28] | 31717ms | 16698K | 32.68/0.8999 | 28.88/0.7891 | 27.82/0.7444 | 27.02/0.8146 | 31.57/0.9185 |

The quantitative comparison results are shown in Table 1. The results indicate that among the LUT-based SR methods, ECNN-L8C8 achieved the highest PSNR and SSIM quality. Compared with the earliest method SR-LUT, ECNN-L8C8 improved PSNR by 1.22dB on the Set5 dataset, and was close to SR-LUT regarding running time and LUT size. It is worth noting that the lower complexity ECNN-L6C6 surpassed all previous LUT-based methods across metrics, except for running slower than EC-LUT-V, thus demonstrating the superiority of ECNN. Compared with ECNN-L8C8, ECNN-L6C6 compromised only a small part of SR quality, but gained twice the running speed, showing better overall performance. Among all the LUT-based methods, ECNN-L4C4 had the shortest running time, yet experienced a more significant decrease in SR quality.

Overall, the single LUT size of ECNN is at the KB level. ECNN-L4C4 and ECNN-L6C6 even have a total LUT size of less than 1MB. This reduced LUT size not only minimizes storage space utilization but also enables the KB-level LUT to be seamlessly loaded into the L1 cache, greatly enhancing the cache hit rate and subsequently improving inference speed. Moreover, despite having a smaller RF, ECNN-L6C6 and ECNN-L8C8 exhibit superior SR quality compared to RCLUT, demonstrating the superiority of a multi-channel approach. Additionally, all of our methods have faster inference speed than RCLUT, indicating that the strategy of utilizing multiple small-sized convolution modules instead of a single large-sized one is indeed effective.

## 4.3 Qualitative Comparison

Figure 6 presents a visual comparison of several LUT-based SR methods with the ground truth (GT). Since RCLUT did not provide a trained model, its results are omitted from the figure. Overall, ECNN-L8C8 achieved a considerable quality improvement, and the

**Table 2: Ablation study on quantization strategy. QAT denotes the proposed fine-tune strategy (quantization of LUT indices). PTQ represents the quantization of data stored in LUT. DQ denotes direct quantization without fine-tune.**

| Method | QAT | PTQ | DQ | Set5 | Set14 | B100 | Urban100 | Manga109 |
|---|---|---|---|---|---|---|---|---|
| L4C4 | | | | 30.02 | 27.23 | 26.68 | 24.10 | 27.14 |
| | ✓ | | | 30.01 | 27.21 | 26.67 | 24.09 | 27.10 |
| | ✓ | ✓ | | 29.99 | 27.19 | 26.67 | 24.06 | 27.07 |
| | | ✓ | ✓ | 29.97 | 27.20 | 26.65 | 24.07 | 27.11 |
| L6C6 | | | | 30.76 | 27.72 | 26.97 | 24.60 | 28.16 |
| | ✓ | | | 30.75 | 27.70 | 26.96 | 24.59 | 28.12 |
| | ✓ | ✓ | | 30.75 | 27.70 | 26.96 | 24.58 | 28.12 |
| | | ✓ | ✓ | 30.69 | 27.66 | 26.93 | 24.52 | 27.99 |
| L8C8 | | | | 31.08 | 27.93 | 27.09 | 24.84 | 28.64 |
| | ✓ | | | 31.07 | 27.91 | 27.08 | 24.82 | 28.59 |
| | ✓ | ✓ | | 31.06 | 27.91 | 27.08 | 24.82 | 28.59 |
| | | ✓ | ✓ | 30.71 | 27.69 | 26.93 | 24.55 | 28.09 |

SR results were clearly sharper than other methods. While ECNN-L6C6 exhibited a slight reduction in SR quality, it still outperformed other methods.

Specifically, comparing the first row, only ECNN-L6C6 and ECNN-L8C8 produced a good transition at the circular arc in the lower left corner of the mirror, while all other methods produced jagged edges. In the second and third rows, all other methods produced oblique stripe-like artifacts, whereas ECNN-L6C6 and ECNN-L8C8 effectively suppressed such artifacts. In the fourth row, there are two obvious black lines in the picture, which other methods restored as wavy lines, and there are also some reverse stripes in the middle.

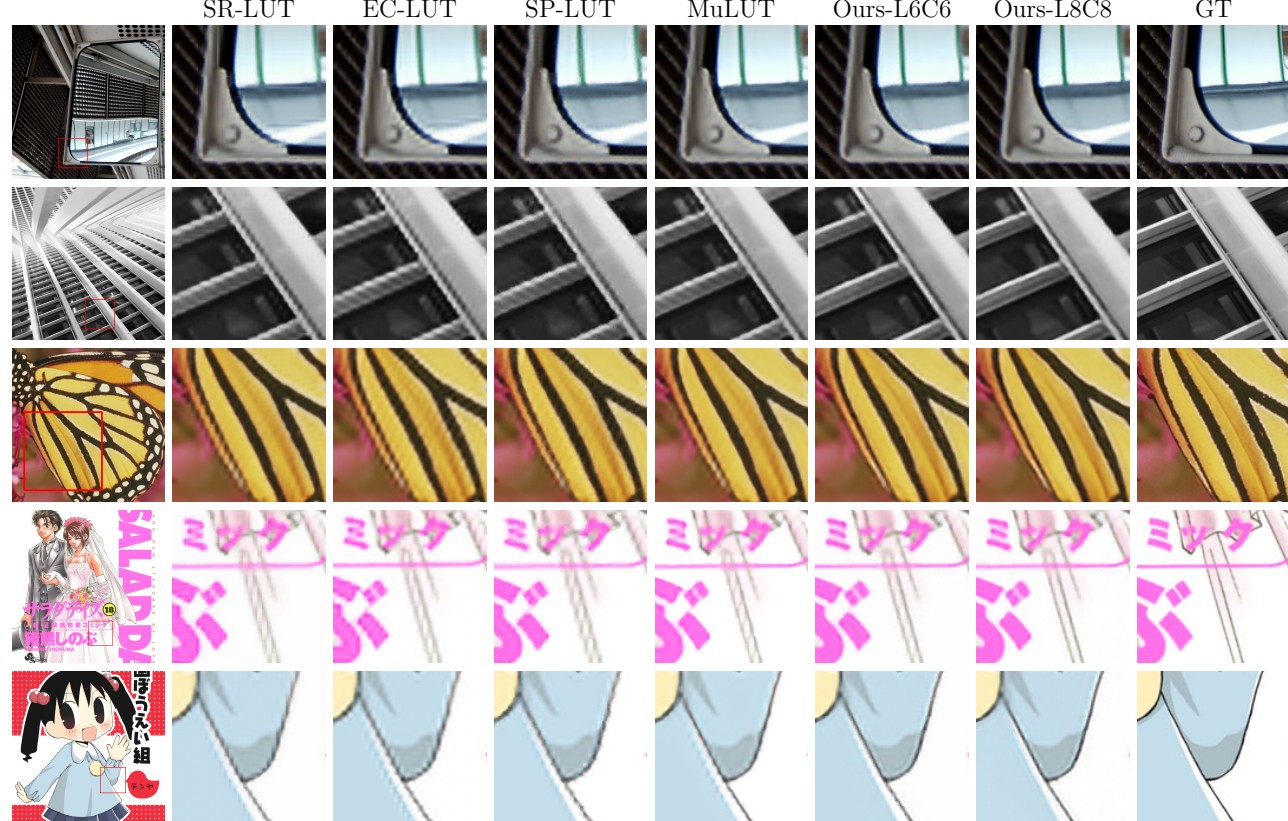

**Figure 6: Visual comparison for ×4 SR on benchmark datasets. The results indicate that ECNN-L6C6 and ECNN-L8C8 exhibit significant visual improvements, including fewer artifacts and jagged edges, as well as sharper edges.**

Our ECNN-L8C8 restored the lines effectively. In the last row, the white highlight on the edge of the clothes, ECNN generated a sharp edge, whereas other methods produced jagged edges. These results show that ECNN can better restore the high-frequency details of the image, while suppressing artifacts, and produce better visual effects.

## 4.4 Ablation Study

**The effectiveness of proposed quantization strategy.** We conducted ablation experiments on ECNN models with different complexities under various quantization strategies. The experimental results in Table 2 demonstrate the effectiveness of the quantization strategy proposed in this paper from two perspectives. Firstly, after undergoing both the QAT and PTQ stages (our strategy), the performance degradation of the three models is minimal, virtually negligible. Secondly, directly incorporating quantization nodes during the training process (a strategy employed in previous studies) has a minimal impact on the ECNN-L4C4 model, whereas the performance degradation of the ECNN-L8C8 model is significant. This indicates that when there are a large number of cascaded LUTs, the DQ strategy fails to achieve optimal performance, whereas our strategy remains effective regardless.

**Table 3: Ablation study on the runtime performance of EC-NNs with and without PTQ.**

| ECNN-L4C4 | | ECNN-L6C6 | | ECNN-L8C8 | |
|---|---|---|---|---|---|
| PTQ | w/o PTQ | PTQ | w/o PTQ | PTQ | w/o PTQ |
| 31ms | 32ms | 61ms | 70ms | 119ms | 134ms |

**Table 4: Ablation study on different bit width quantization for LUT, where Float32 indicates no quantization, and other numbers denote the number of bits used for quantization. We choose 12 bits as the optimal setting considering performance and LUT size.**

| Bit | Size | Set5 | Set14 | B100 | Urban100 | Manga109 |
|---|---|---|---|---|---|---|
| 8 | 1.029MB | 29.94 | 27.19 | 26.48 | 24.38 | 27.84 |
| 9 | 1.157MB | 30.64 | 27.65 | 26.83 | 24.67 | 28.35 |
| 10 | 1.286MB | 30.99 | 27.86 | 27.04 | 24.80 | 28.55 |
| 11 | 1.415MB | 31.04 | 27.89 | 27.06 | 24.81 | 28.57 |
| 12 (Ours) | 1.543MB | 31.06 | 27.91 | 27.08 | 24.82 | 28.59 |
| Float32 | 4.116MB | 31.07 | 27.91 | 27.08 | 24.82 | 28.59 |

VanillaCNN-L6C6

ECNN-L6C6

**Figure 7: Visualization of learned feature maps for vanilla CNN and ECNN.**

**The impact of PTQ quantization bit width.** Although using 16-bit low-precision floating-point numbers as the storage format for LUT data is acceptable, as demonstrated by SP-LUT, fixed-point numbers still excel in computational efficiency. Therefore, it remains necessary to quantize this portion of data to further enhance performance. According to Table 3, PTQ effectively offers acceleration. Meanwhile, as shown in Table 4, when quantized to 12 bits, the model performance hardly suffers any decrease. But as the quantization level is reduced below 10 bits, the model performance experiences a significant drop. Notably, when quantized to 8 bits, the PSNR decreases by 1.13dB on the Set5 dataset. To balance both LUT size and SR quality, we opted to quantize to 12 bits.

## 4.5 Comparison with Vanilla CNN

To compare the performance difference between the proposed ECNN and the ordinary convolutional neural network, experiments were conducted under the same conditions of layer number, channel number and equivalent convolution kernel size. As shown in Table 5, the three ECNN models with different complexities all surpassed the ordinary CNN network in PSNR quality. This is because the ordinary convolution operation is simply a linear mapping that multiplies the weights and inputs and then sums them up, and its nonlinear mapping ability depends on the nonlinear activation function. ECNN first performs nonlinear mapping on the input data and then sums them up, which can approximate a more abstract representation of the latent knowledge. This is similar to the idea of Network In Network [18]. As shown in Figure 7, compared with vanilla CNN, ECNN extracts more high-frequency signals in both the first and last layers. This explains why ECNN models

**Table 5: Comparison of efficiency between vanilla CNN and ECNN. Both have consistent equivalent RF, as well as the same number of convolutional layers and hidden layer channels.**

| Method | Set5 | Set14 | B100 | Urban100 | Manga109 |
|--------|------|-------|------|----------|----------|
| Vanilla-L4C4 | 29.19 | 26.57 | 26.33 | 23.55 | 25.67 |
| ECNN-L4C4 | 29.99 | 27.19 | 26.67 | 24.06 | 27.07 |
| Vanilla-L6C6 | 29.41 | 26.70 | 26.42 | 23.66 | 25.84 |
| ECNN-L6C6 | 30.75 | 27.7 | 26.96 | 24.58 | 28.12 |
| Vanilla-L8C8 | 29.44 | 26.73 | 26.43 | 23.68 | 25.86 |
| ECNN-L8C8 | 31.06 | 27.91 | 27.08 | 24.82 | 28.59 |

with the same complexity perform better than vanilla CNNs. In addition, ECNN converted to LUT completely avoids multiplication and floating-point operations, and has more advantages in computational efficiency than ordinary convolution.

## 5 CONCLUSION

In this paper, we propose the complete expanded convolution, which can be used to build ECNN models like vanilla convolution. After the ECNN training, we add quantization nodes to fine-tune the model and generate the equivalent LUT for inference stage acceleration. Our method only uses a single value as an index, which greatly reduces the size of each LUT and increases the cache hit rate. More importantly, we do not need the interpolation and rotation operations, which significantly improves the inference speed. In addition, ECNN increases the receptive field and enhances the mapping ability by using multi-channel multi-layer convolution, significantly improving SR quality. Experiments show that our models offer superior SR quality, faster inference, or smaller LUT volume compared to comparable LUT-based methods. In the future, we will explore arbitrary-scale SR methods based on ECNN to improve the practicality of ECNN further.

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

Received 20 February 2007; revised 12 March 2009; accepted 5 June 2009

