# OpenReview forum: "Expanded Convolutional Neural Network Based Look-Up Tables for High Efficient Single-Image Super-Resolution"
_acmmm.org/ACMMM/2024/Conference — MM2024 Poster_

### Official Review · Reviewer_w2WL · 2024-05-15

**Rating:** 5
**Confidence:** 3

**Summary:**

The authors propose an expanded CNN (ECNN) based look-up table for highly efficient single-image super-resolution. The authors extend feature fusion to the feature channel dimension to enhance mapping ability. In addition, the authors reduce the number of single indexed pixels to just one, reducing the LUT size from the MB level to the KB level. With the improvements, the authors stack expanded convolutional layers to form an ECNN, with each layer convertible to LUTs during inference.

**Strengths:**

1) The authors propose ECNN, a novel LUT-based approach for SR task, by reducing the number of single indexed pixels to just one.
2) The authors extend feature fusion to the feature channel dimension to enhance mapping ability.
3) The proposed ECNN improves the overall performance of the upper limit of LUT based methods.

**Limitations:**

1) Receptive field is important for the SR performance. The authors use expanded convolution with input window size 1x1. How do the authors implicitly improve the receptive field? Please explain, because 1x1 receptive field cannot result in good performance.
2) It is not clear how the authors up-sample the feature maps in the final layer. Fig. 2 and the explanation in Sec. 3.2 are confusing. How do the authors aggregate the feature maps along the channel axis to compress back to the low-dimensional RGB space?
3) The authors choose to quantize the LUTs to 12bit. However, since there is no 12-bit data type, is it better to quantize to 16 bits?

**Suitability:**

2

---

### Official Review · Reviewer_CerW · 2024-05-24

**Rating:** 3
**Confidence:** 1

**Summary:**

The paper proposes a novel super-resolution technique called expanded convolutional neural network (ECNN) to address the limitations of existing methods. The ECNN approach extends feature fusion to enhance mapping ability and reduces the LUT size, improving cache hit rates. The experiments demonstrate that the ECNN outperforms previous methods in terms of speed and LUT volume while achieving state-of-the-art performance in super-resolution quality. Overall, the paper presents a promising solution to the challenges faced inefficient super-resolution approaches.

**Strengths:**

- The proposed approaches reduce the LUT size effectively.
- The experiment results show that ECNN is more effective than previous approaches.

**Limitations:**

- More state-of-the-art approaches should be compared to as CARN (2016) , FSRCNN (2018) and RRDB (2018) are proposed quite a long ago.
- The paper would benefit from additional refinement and expanded content to provide a more thorough explanation of the ECNN.
- Many expressions require additional polish.
e.g. At this point, the size of LUTs has been significantly reduced from the GB or MB levels to the KB levels.

**Suitability:**

2

---

### Official Review · Reviewer_sovc · 2024-05-25

**Rating:** 4
**Confidence:** 3

**Summary:**

This paper presents a novel expanded convolutional neural network (ECNN) that enhances feature fusion in the channel dimension and reduces the number of single indexed pixels, significantly decreasing the LUT size and improving inference speed. Experimental results demonstrate that this method achieves state-of-the-art performance in both speed and LUT volume under comparable SR quality conditions.

**Strengths:**

1.This paper is well-written with a complete structure, and I did not observe any significant grammatical or spelling errors.

2.The authors have taken a practical approach, inspired by reference [30], identifying issues with SR-LUT, and proposing a new method, ECNN, to address the limitations related to "SR quality".

3.The introduction is logically structured. The authors progressively approach the core problem from an application perspective.

4.The authors have validated the rationale of their quantization technique through extensive testing and ablation studies, confirming the effectiveness of ECNN.

**Limitations:**

1.Despite the many strengths of this paper, there are areas for improvement. Firstly, regarding motivation, from line 125, the authors introduce the main problem they are addressing but fail to explain or justify these challenges. For example, the authors mention that the rotation technique in EC-LUT is "without conferring any significant advantage." However, in my understanding, rotation is a data augmentation technique that helps the model learn features from different directions, which is independent of the network structure itself. Are the authors implying that in the context of LUT methods, it increases overhead? I did not find an explanation or validation of the negative impacts of rotation in the paper. Additionally, I noticed that by line 281, the discussion of rotation changes from offering no advantage to having a "negative impact," which further confused me.

2.This paper’s references to prior work enhance its technical foundation. However, as a reviewer, I do not recommend dedicating too much space to describing previous work. The authors state in the abstract that this work is an incremental improvement based on EC-LUT, which naturally leads readers to look for distinctions between the two. I can see the authors' efforts in this regard, as they have used several paragraphs in Sections 1, 2, and 3 to introduce EC-LUT. After reading the paper, even without a detailed understanding of EC-LUT, I have a clear grasp of its motivation. The balance between receptive field size (affecting performance) and LUT size is crucial for LUT-based SR techniques, and the introduction of EC mitigates the performance impact of reducing the receptive field to 2x2. Since the motivation for previous work is already well established, it seems that an excessive introduction of prior work reduces the space available to highlight the novelty of the proposed method.

3.The paper mentions reducing the convolution size to 1x1 to decrease the LUT size to the KB level. Based on my knowledge, 1x1 convolutions are typically used for dimensionality reduction of feature maps and have weaker feature extraction capabilities compared to larger kernels, which is why previous LUT work used 3x3 or 2x2. The technique illustrated in Figure 3 seems similar to the PixelShuffle upsampling method, but with the deconvolution kernel replaced by LUT and multiple output channels summed. This makes ECNN's EC part resemble upsampling with pixel-level fusion, which indeed reduces LUT size, but what are the trade-offs? I am uncertain about the reliability and effectiveness of this approach.

4.The advantage of LUT-based SR methods lies in converting computational tasks of CNNs into LUT queries. This paper appears to retain the complete CNN and replaces a single convolution process with LUT. This significant change requires more detailed text to describe its motivation.

5.Although I am reluctant to overemphasize innovation, as it is difficult to quantify, there are concerns in this area. The key "E" (Expanded) in the proposed ECNN comes from another work in the same field (EC-LUT). Despite potential incremental improvements, the lack of a sufficiently distinct differentiation in the algorithm name from "Expanded Convolution" used previously is worrisome.

**Suitability:**

2

---

### Meta-Review · Area_Chair_F6JS · 2024-06-30

**Recommendation:** Accept (Poster)
**Confidence:** 3

**Metareview:**

The final ratings are 1 weak accept, 1 borderline accept, and 1 borderline reject. The reviewers found that the paper is well written and the results are promising. Overall, the paper seems to have merits for acceptance. Nevertheless, there is a recurring question on clarification of the 1x1 conv vs. receptive field; it is suggested that the authors elaborate to resolve the issue.